Journal of Machine Learning Research 23 (2022) 1-9         Submitted 1/21; Revised 5/22; Published 9/22

# Combining AI-based Biomarkers for PDAC Survival

**Lisa Hensens**                            LISA.HENSENS@RU.NL
*Department of Pathology, Radboud University Medical Center, 6500 HB Nijmegen, The Netherlands.*

**Sergio Sabroso-Lasa**                      SSABROSO@CNIO.ES
*Spanish National Cancer Research Center, Genetic & Molecular Epidemiologi Group, CIBERONC, Madrid, 28029, Spain.*

**Caroline Verbeke**                      C.S.VERBEKE@MEDISIN.UIO.NL
*Department of Pathology, Oslo University Hospital, Oslo, Norway.*

**Nuria Malats**                             NMALATS@CNIO.ES
*Spanish National Cancer Research Center, Genetic & Molecular Epidemiologi Group, CIBERONC, Madrid, 28029, Spain.*

**ThePanGenEU consortium**                  NMALATS@CNIO.ES
*Spanish National Cancer Research Center, Genetic & Molecular Epidemiologi Group, CIBERONC, Madrid, 28029, Spain.*

**Geert Litjens**                        GEERT.LITJENS@RADBOUDUMC.NL
*Department of Pathology, Radboud University Medical Center, 6500 HB Nijmegen, The Netherlands.*

**Pierpaolo Vendittelli**             PIERPAOLO.VENDITTELLI@RADBOUDUMC.NL
*Department of Pathology, Radboud University Medical Center, 6500 HB Nijmegen, The Netherlands.*

**Editor:** My editor

## Abstract

Pancreatic ductal adenocarcinoma (PDAC) remains one of the deadliest cancers due to late detection and limited treatment response. This study investigates the prognostic value of combining multiple AI-based image biomarkers—tumor-stroma ratio (TSR), mitosis density, stromal cell density (via HoVer-Net), tumor-to-tissue ratio from histopathological whole-slide images (WSIs) for survival prediction in resected PDAC patients. A multi-tissue segmentation model was developed to generate tissue masks for downstream biomarker extraction. Using logistic and Cox regression models, both univariate and multivariate survival analyses were performed across four datasets. Results show that while combining biomarkers did not outperform single-biomarker models (notably TSR), mitosis density showed consistent statistical significance and may serve as a valuable prognostic feature.

**Keywords:** Segmentation, Deep learning, Pancreatic ductal adenocarcinoma, AI-based biomarkers, survival analysis

## 1 Introduction

Pancreatic ductal adenocarcinoma (PDAC) is a highly lethal malignancy with the lowest 5-year survival rate among major cancers (Siegel et al. (2023)). Its poor prognosis is largely

due to late-stage diagnosis and limited treatment efficacy (Latenstein et al. (2020)). Advances in digital pathology and artificial intelligence (AI) have enabled the extraction of image-based biomarkers directly from histopathological whole-slide images (WSIs), offering new potential for prognostication.

Several individual biomarkers, such as the tumor-stroma ratio (TSR) (Li et al. (2020); Vendittelli et al. (2024)), tumor-to-tissue ratio (TTR), mitotic figure density (Tellez et al. (2018)), and stromal cell density Graham et al. (2019); Sántha et al. (2021), have demonstrated prognostic value in isolation. However, the combined impact of these features on overall survival in PDAC patients remains unclear. While multiple-instance learning (MIL) and other weakly supervised deep learning methods have recently become the dominant paradigm for WSI-based prognosis prediction, these end-to-end models typically operate as "black boxes" and provide limited insight into the morphological features driving their predictions. In contrast, this study deliberately adopts an interpretable, multi-step approach that quantifies human-understandable image biomarkers, allowing for biological interpretation and clinical validation. Specifically, we investigate whether combining AI-quantified biomarkers—tumor-stroma ratio (TSR), tumor-to-tissue ratio (TTR), mitotic cell density, and stromal cell density—can improve survival prediction in PDAC. These biomarkers were extracted from WSIs using dedicated deep learning pipelines for tissue segmentation, nucleus classification, and mitosis detection, and their prognostic value was evaluated through univariate and multivariate survival analyses using both logistic and Cox regression models.

## 2 Data

Table 1: Overview of the datasets used in this study.

| Source | Patients | Slides |
|---|---|---|
| Radboudumc | 122 | 2500 |
| TCGA | 161 | 187 |
| CPTAC | 128 | 128 |
| pain PanGenEU | 75 | 75 |

Four independent datasets were considered in this study (Table 1): Radboudumc, TCGA-PAAD, CPTAC-PDA, and PanGenEU. Data were anonymized, and informed consent was waived where applicable. For patients with multiple slides, the most representative slide, typically the one containing the most amount of tumor, was selected.

We collected clinical variables including vital status and survival time (days between diagnosis and death or last follow-up).

## 3 Methods

Our pipeline consists of: (1) tumor microenvironment segmentation, (2) nuclear segmentation/classification, (3) mitosis detection, and (4) image-based biomarker extraction.

**Tumor microenvironment segmentation** A U-Net Ronneberger et al. (2015) was trained to perform multi-tissue segmentation of H&E-stained pancreatic histopathology

slides. The network was trained on a multicentric private dataset comprising 162 resected PDAC cases. Patches of size $512 \times 512$ pixels were extracted at a resolution of 1.0 micron/pixel. Model training was conducted using five-fold cross-validation, ensuring that cases were split at the patient level to prevent data leakage across folds.

Ground-truth annotations were derived in a semi-automated fashion. An epithelium segmentation network previously trained on PDAC cases Vendittelli et al. (2024) was used to generate annotations for both tumor and healthy epithelium. The remaining tissue components, stroma, fat, mucus, muscle, and granulocytes, were annotated using a multi-tissue segmentation model pretrained on colorectal cancer tissue Bokhorst et al. (2021).

Based on the network's predictions, we defined the tumor bulk as a composite region encompassing both the tumor epithelium and the surrounding stroma. This region was essential for the computation of all downstream biomarkers, including TSR, mitosis density, and TTR. To approximate the spatial extent of the tumor bulk, we generated a convex hull around the segmented tumor epithelium using the alphahull algorithm, with an empirically determined alpha parameter of 0.038.

**Segmentation evaluation:** Model performance was assessed using the Dice coefficient. On internal five-fold cross-validation, the network achieved the following median Dice scores (interquartile range, IQR): 0.67 (0.19) for tumor epithelium, 0.58 (0.40) for healthy epithelium, 0.85 (0.08) for stroma, 0.90 (0.15) for fat, 0.61 (0.17) for mucus, 0.67 (0.16) for muscle, and 0.80 (0.18) for granulocytes. The convex-hull-based tumor bulk segmentation achieved a median Dice score of 0.7 (with a standard deviation of 0.27).

To assess generalization, the model was evaluated on a subset of 33 TCGA-PAAD cases that an expert pathologist independently annotated. On this external test set, the model achieved Dice scores of 0.72 (0.17) for tumor epithelium and 0.81 (0.20) for tumor bulk, confirming good generalization performances.

A visual example of multi-tissue segmentation is shown in Figure 2.

Table 2: Segmentation performance on TCGA-PAAD dataset.

| Structure | Dice (External) |
|---|---|
| Tumor epithelium | 0.72 |
| Tumor bulk | 0.81 |

**Nuclear Segmentation and Classification**  Inside the segmented tumor, HoVer-Net (Graham et al. (2019)) was used to segment and classify nuclei into epithelial, inflammatory, stromal, or miscellaneous. The model was applied on $256 \times 256$ pixel tumor patches at $40\times$ magnification.

**Mitotic Figure Detection**  A mitosis detector (Tellez et al. (2018)) was applied to the slides and then masked with the tumor bulk region. We computed mitotic density as the number of mitoses per $mm^2$ of tumor area, retaining only high-confidence predictions. A visual example of the mitosis algorithm can be seen in Figure 3.

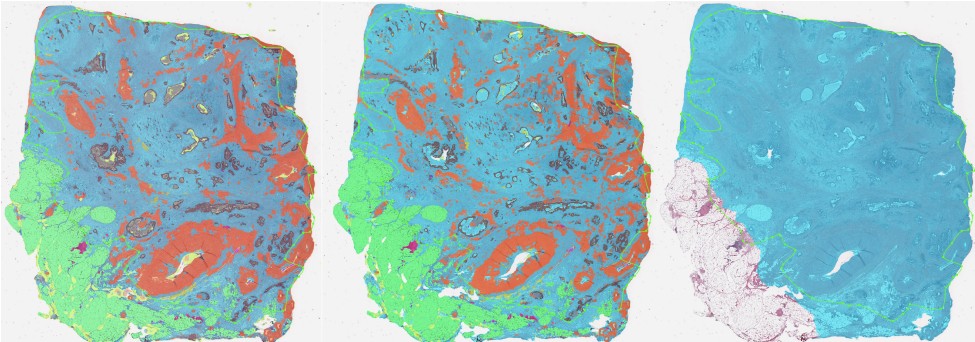

Figure 1: Example of Multi tissue segmentation output and convex hull. First is the output of the network, middle is the ground truth, and right represents the output of the alpha hull algorithm. Key components are tumor epithelium (in dark) and stroma (in blue).

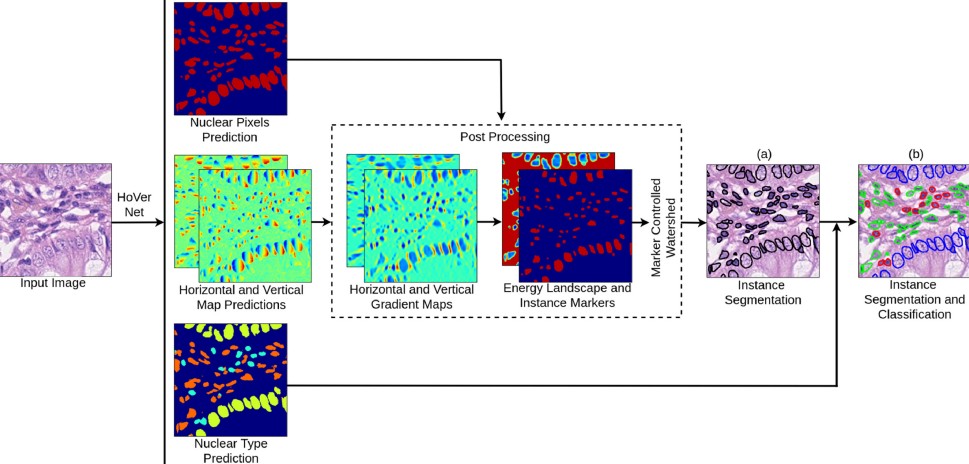

Figure 2: Overview of Hovernet as described by the authors.The different colors represent different types of nuclei.

## 3.1 Image-Based Biomarkers

From the output of the tumor microenvironment segmentation network, a tumor bulk region and its internal tissue structures were derived, as described in the section above. Based on these segmentations, four independent biomarkers were extracted: the TSR, defined as the ratio of tumor epithelium to surrounding stroma; the mitosis density, calculated as the number of mitotic figures per $mm^2$ of tumor bulk area; the nuclear features, quantified as the cell-type-specific nuclear density (cells per $mm^2$) within the tumor region; and the TTR, defined as the ratio of tumor bulk area to total tissue area.

## 4 Results

A correlation analysis was performed between these features and overall survival in both univariate and multivariate settings. Additionally, when multivariate analysis was performed,

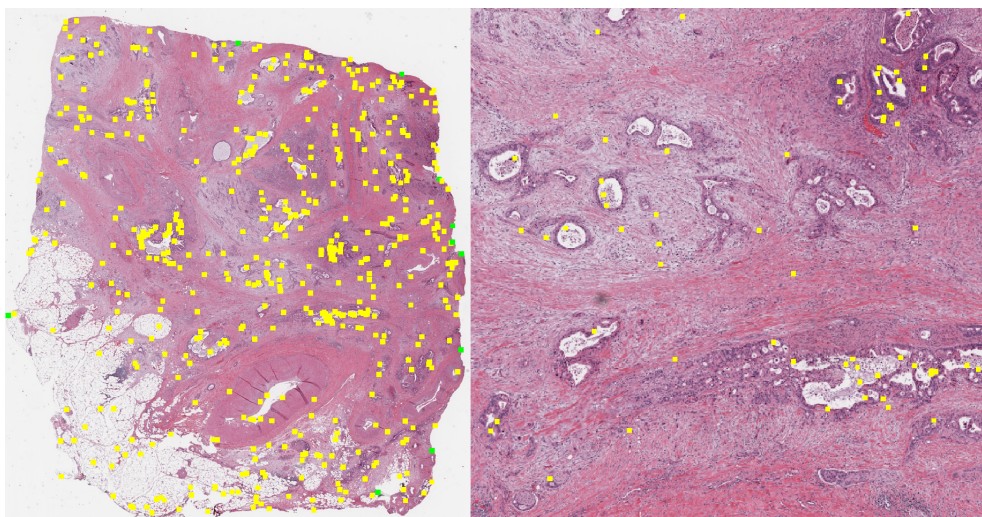

Figure 3: Example of mitosis algorithm on a TCGA-PAAD case. The detected mitotic figures are first computed inside the slide, then masked with the tumor bulk mask.

several clinical features were also taken into account. Selected clinical features included age and gender. As a clinical endpoint, vital status and the days to survival were also considered.

### 4.1 Univariate Survival Analysis

Table 3: Univariate Cox regression (n=368). Bold p-values are significant ($p < 0.05$). *Normalized features.

| Biomarker | C-index | HR (95% CI) | p-value |
|-----------|---------|-------------|---------|
| TTR | 0.54 | 1.79 (1.14–2.79) | **0.01** |
| Mitosis density* | 0.51 | 1.14 (1.00–1.31) | **0.05** |
| Stromal cell density* | 0.52 | 0.96 (0.84–1.08) | 0.47 |

The univariate Cox regression analysis (Table 3) identified Tumor-to-Tissue Ratio (TTR) as a significant prognostic factor for overall survival (HR=1.79, 95% CI: 1.14–2.79, p=0.01), with a concordance index (C-index) of 0.54. This suggests that higher TTR values are associated with an increased hazard of death, although the relatively modest C-index indicates limited discriminative ability when used in isolation. Mitosis density also reached statistical significance at the threshold level (p=0.05), with an HR of 1.14 (95% CI: 1.00–1.31), indicating a weak yet potentially relevant association with poorer survival. In contrast, stromal cell density did not show a significant relationship with survival (HR = 0.96, p=0.47), and its C-index (0.52) was close to random prediction. The univariate logistic regression analysis (Table 4) examined the predictive performance of individual biomarkers for binary survival outcomes at 6, 12, and 18 months. TSR consistently showed the highest AUC values across all time points, peaking at 12 months (AUC =$0.63 \pm 0.06$), suggesting mod-

Table 4: Univariate logistic regression (5-fold CV), AUC ± std. *Normalized features.

| Biomarker | 6 mo | 12 mo | 18 mo |
|---|---|---|---|
| TTR | 0.57 ± 0.12 | 0.56 ± 0.04 | 0.54 ± 0.05 |
| TSR | 0.61 ± 0.11 | 0.63 ± 0.06 | 0.57 ± 0.07 |
| Mitosis density* | 0.54 ± 0.06 | 0.52 ± 0.08 | 0.53 ± 0.07 |
| Stromal cell density* | 0.61 ± 0.06 | 0.51 ± 0.07 | 0.56 ± 0.07 |

erate discriminative ability. TTR achieved slightly lower AUCs, ranging from $0.54 \pm 0.05$ to $0.57 \pm 0.12$, consistent with its limited performance observed in the Cox analysis. Mitosis density and stromal cell density exhibited poor discrimination (AUCs generally close to 0.5), indicating limited utility as standalone classifiers of short-term survival. Overall, these findings underscore the limited predictive power of individual biomarkers and suggest that multivariate or integrative approaches may be required to achieve clinically meaningful prognostic performance.

## 4.2 Multivariate Survival Analysis

Table 5: Multivariate Cox regression with/without clinical features (n=368).

| Model | HR (95% CI) | p-value | C-index |
|---|---|---|---|
| TTR | 1.84 (1.08–2.95) | **0.01** | |
| Mitosis density | 1.18 (1.03–1.35) | **0.02** | 0.54 |
| TTR | 1.83 (1.14–2.94) | **0.01** | |
| Mitosis density | 1.17 (1.03–1.34) | **0.02** | |
| Age | 1.94 (0.56–6.65) | 0.29 | 0.55 |
| Gender | 0.97 (0.75–1.24) | 0.79 | |

The multivariate Cox regression models (Table 5) assessed the independent prognostic value of TTR and mitosis density, both with and without the inclusion of clinical covariates. The stromal cell density biomarker was not included in the multivariate analysis as its HR was very close to 1, representing poor correlation with survival. In the model excluding clinical variables, both TTR (HR = 1.84, 95% CI: 1.08–2.95, p=0.01) and mitosis density (HR = 1.18, 95% CI: 1.03–1.35, p=0.02) were significantly associated with overall survival. When adjusted for clinical factors, the prognostic significance of both biomarkers was retained, with nearly identical hazard ratios and slightly attenuated p-values (TTR: HR = 1.83, p=0.01; mitosis density: HR = 1.17, p=0.02).

The addition of clinical features only marginally improved the model's discriminative performance, increasing the C-index from 0.54 to 0.55. This indicates that while TTR and mitosis density are statistically significant independent predictors of survival, their overall contribution to risk stratification remains modest when considered within a multivariate framework.

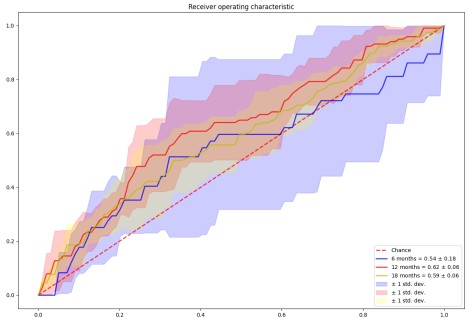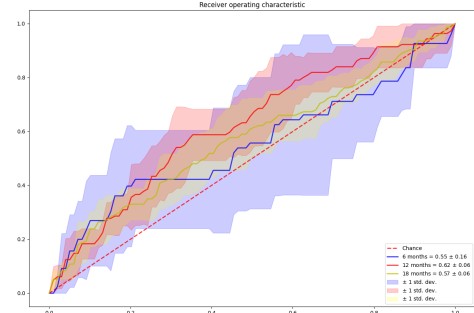

(a) Logistic regression without clinical variable (n=368)   (b) Logistic regression with clinical variable (n=368)

Table 6: AUC (95% CI) for training set.

| Model | 6 mo | 12 mo | 18 mo |
|---|---|---|---|
| With clinical | $0.55 \pm 0.16$ | $0.62 \pm 0.06$ | $0.57 \pm 0.06$ |
| Without clinical | $0.54 \pm 0.18$ | $0.62 \pm 0.06$ | $0.59 \pm 0.06$ |

## 4.3 External Validation

The final models were validated on an independent test set from the PanGenEU cohort (n=74). Logistic regression models trained on the combined training dataset were evaluated at three clinically relevant timepoints. AUC scores were calculated for each model, with and without clinical covariates.

Table 7: AUC (95% CI) for external test set (PanGenEU).

| Model | 6 mo | 12 mo | 18 mo |
|---|---|---|---|
| With clinical | 0.58 (0.39–0.78) | 0.46 (0.31–0.61) | 0.43 (0.30–0.56) |
| Without clinical | 0.58 (0.37–0.81) | 0.43 (0.29–0.57) | 0.43 (0.29–0.56) |

External evaluation showed moderate predictive performance for short-term survival, while the ability to discriminate longer-term outcomes was limited, with AUC values nearing baseline and no statistically significant differences observed. Nonetheless, mitotic density and TTR remained the most consistent predictors across datasets, highlighting their potential relevance as generalizable image-based biomarkers for PDAC prognosis.

## 5 Discussion and Conclusion

This study evaluated the prognostic value of AI-derived image biomarkers from WSIs in PDAC. Among all features, mitotic density and TTR were the most consistent and statistically significant predictors of survival. The linear models used here offer interpretability, although their predictive performance remained modest.

A key prerequisite for biomarker reliability is accurate tissue segmentation. Our multi-tissue segmentation model achieved solid performance on a held-out test set, with Dice scores of 0.79 for tumor epithelium and 0.81 for tumor bulk. These results support the robustness of our pipeline and suggest that the extracted features are based on reliable spatial annotations. The visual segmentation results further confirm the network's ability to delineate complex tumor microenvironments.

Although combining features in multivariate models did not outperform single-biomarker models, integrated models may still be useful for interpretability and risk stratification. Notably, adding clinical features provided only marginal gains in discriminative power, highlighting the independent prognostic relevance of mitosis density and TTR. The modest predictive performance observed in this work reflects both the biological complexity of PDAC and potential technical limitations. In particular domain shifts between cohorts likely contributed to reduced generalization in the external validation.

Limitations of this study include moderate cohort sizes, class imbalance, and reliance on linear models. Additionally, external validation showed reduced generalization, likely due to cohort heterogeneity and domain shifts. Future work should explore non-linear modeling approaches (e.g., deep survival networks), spatial modeling of biomarker patterns, and the integration of molecular or genomic data with image-based features. Future studies would include direct comparisons with MIL-based survival models would also help contextualize the trade-off between interpretability and raw predictive performance.

In conclusion, mitotic density and TTR are promising, generalizable biomarkers for PDAC prognosis. Our findings support the integration of AI-based segmentation and feature extraction pipelines into computational pathology frameworks for survival modeling and clinical research.

## Acknowledgments and Disclosure of Funding

This project has received funding from the European Union's Horizon 2020 research and innovation programme under grant agreement No. 101016851 (PANCAIM). Additional funding support was provided by: EU-FP7-HEALTH (Grant/Award Numbers: 256974-EPC-TM-Net, 259737-CANCERALIA); EU-6FP Integrated Project (Grant/Award Number: 018771-MOLDIAG-PACA); Fondo de Investigaciones Sanitarias (FIS), Instituto de Salud Carlos III, Spain (Grant/Award Numbers: PI11/01542, PI0902102, PI12/01635, PI12/00815, PI15/01573, PI18/01347, PI21/00495).

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
