# OpenReview forum: "Combining AI-based Biomarkers for PDAC Survival"
_MICCAI.org/2025/Workshop/COMPAYL — COMPAYL 2025_

### Official Review · Reviewer_EpWD · 2025-07-12
**Interesting problem, deep evaluation, inconclusive results**

**Rating:** 4
**Confidence:** 4

**Review:**

### Summary

The authors study an interesting problem of predicting prognosis for Pancreatic ductal adenocarcinoma.
Their solution uses a combination of existing methodologies in computational pathology: tissue segmentation, nucleus detection and mitosis classification. Biomarkers are produced on top of the deep learning outputs.

### Strenght
- The problem is interesting and well-defined
- The datasets are large
- The model is sound
- Models are well evaluated.

### Weaknesses
- Overall performance is very weak; it seems like the biomarker does not correlate with the prognosis.
- Although negative results should be appreciated since they help to rule out failure models, the conclusion of the paper does not rule out anything since the performance is too bad. Still, the effort is appreciated and inspires further investigation.

---

### Official Review · Reviewer_5zVu · 2025-07-13
**An interpretable pipeline for PDAC survival analysis**

**Rating:** 4
**Confidence:** 4

**Review:**

# Summary

The authors propose a multi-step pipeline for predicting survival in pancreatic ductal adenocarcinoma (PDAC). The pipeline consists of four steps: tumor segmentation, nucleus segmentation/classification, mitosis detection, and extracting AI-based image biomarkers. They then perform Cox regression to predict survival based on extracted features. Interestingly, unlike end-to-end deep learning approaches, the individual features themselves are interpretable, e.g. “the ratio of tumor epithelium to surrounding stroma”.

# Strengths

- The authors evaluate not only the performance of the overall pipeline, but also the performance of individual steps.
- The authors perform statistical tests and report p-values for their findings.
- The paper focuses on a PDAC, one of the deadliest cancers with urgent need for better prognostic tools.
- Multiple cohorts are used for evaluation.

# Weaknesses

- There is limited technical novelty in the proposed pipeline.
- The pipeline achieved only very modest predictive performance.
- The authors do not perform p-value correction for multiple testing, e.g. Bonferroni or Chow-Denning, (or do not explain how they perform p-value correction if they did).

# Detailed comments

- The introduction lacks a statement about the motivation of the study. Specifically, the reader will want to know why this paper studies the use of combined biomarkers as opposed to using end-to-end weakly supervised whole slide image classification (the latter being the standard approach in the field). From some parts of the manuscript I glean that a motivating reason for this setup was explainability/interpretability, but I think the authors should better explain their motivation in the introduction.
- Table 1: why does the Radboudume dataset have one order of magnitude more slides even though it has similar number of patients as the other datasets?

# Conclusion

While the performance remains modest, I recommend this paper for weak accept because the methodology is sound and the pipeline is evaluated using multiple cohorts. I would probably rate this paper with a “strong accept” if the authors included experiments with end-to-end deep learning approaches (instead of linear combinations of features), as that is the predominant paradigm in the field.

---

### Official Review · Reviewer_tmpi · 2025-07-15
**Exploration of image features towards prognostic predictions**

**Rating:** 3
**Confidence:** 5

**Review:**

The study investigate prognostic potential of image-based biomarkers for patients with pancreatic ductal adenocarcinoma

Pros: Interesting problem

Cons:
- The proposed method do not really use AI to identify prognostic biomarkers, instead it use AI model to extract human defined features from images such as tumor-stroma ratio and then train Cox model for survival prediction. The authors do not proposed any new method instead apply several existing methods to extract image information (this is not necessary cons). However, the method is quite complex, involves lot of pre-processing and the results are not very strong -- and so it is not clear what is the benefit of the proposed approach.
- Given the intensive preprocessing it is not clear how does e.g. the segmentation error translates into survival prediction. This is bit concerning since the segmentation scores are also not so high
- In the time where people use weakly supervised methods such as multiple-instance learning that do not require so extensive data pre-processing as the proposed method the benefits of this work are not so clear (it would help if the authors could better explain motivation of their work)

Summary:
While the study is interesting it needs better justification and explanation on why the proposed work is beneficial.


Which is interesting problem but it involves lot of pre-processing, the results are not very strong and it is not clear what is the benefit of the proposed method over multiple instance learning approach that do not require such intensive data pre-processing
-